# Explosion-Related Polytrauma from Illicit Pyrotechnics: Two Case Reports and a Public Health Perspective

**DOI:** 10.3390/ebj6020031

**Published:** 2025-06-03

**Authors:** Maria Fueth, Simon Bausen, Sonja Verena Schmidt, Felix Reinkemeier, Marius Drysch, Yonca Steubing, Jannik Hinzmann, Marcus Lehnhardt, Elisabete Macedo Santos, Christoph Wallner

**Affiliations:** 1Department of Plastic Surgery, BG University Hospital Bergmannsheil, Ruhr University Bochum, Bürkle-de-la-Camp Platz 1, 44789 Bochum, Germany; simon.bausen@bergmannsheil.de (S.B.);; 2Department of Anesthesiology, BG University Hospital Bergmannsheil, Ruhr University Bochum, Bürkle-de-la-Camp Platz 1, 44789 Bochum, Germany

**Keywords:** firework-related burn injuries, blast trauma, burn wound management, healthcare resource utilization, injury prevention strategies, combined trauma

## Abstract

Firework-related injuries remain a serious public health issue in Germany, especially during New Year’s Eve. While many injuries are minor, the misuse of illegal or homemade fireworks can cause severe trauma resembling military combat injuries and can heavily burden emergency services. Notably, injury rates declined during the COVID-19 firework bans, underscoring the impact of preventive measures. We report two cases of young males with severe injuries from illicit fireworks. The first is a case of a 16-year-old that detonated an illegal Polish firework ball bomb, sustaining 9% total body surface area (TBSA) burns (second- to third-degree), hand fractures, compartment syndrome of the hand, and soft-tissue trauma. He underwent multiple surgeries, including fasciotomy, osteosynthesis, and skin grafting. The other case presented is a 19-year-old man who was injured by a homemade device made of bundled firecrackers, suffering deep facial and bilateral hand burns. He required prolonged ventilation, surgical debridement, and treatment with Kerecis^®^ fish skin and Epicite^®^ dressings. Both required intensive ICU care, interdisciplinary management, and lengthy rehabilitation. Total hospital costs amounted to €58,459.52 and €94,230.23, respectively, as calculated according to the standardized German DRG. These cases illustrate the devastating impact of illegal fireworks. The devastating consequences of explosive trauma are often difficult to treat and may lead to long-term functional and psychological impairments. Prevention through public education, stricter regulations, and preparedness is essential. Pandemic-era injury reductions support sustained policy efforts.

## 1. Introduction

Fireworks are an integral part of German New Year’s Eve festivities, but they also cause numerous injuries each year. These injuries commonly include burns, soft-tissue lacerations, and fractures, with hands and eyes, in particular, being often affected [1,2]. While many incidents are minor, severe or even fatal outcomes occur repeatedly, usually when high-powered fireworks are used improperly or illegally [1]. In a recent New Year period, the Unfallkrankenhaus Berlin (UKB) treated 50 patients with firework-related injuries (many of them children); several required partial or complete finger amputations, numerous suffered burns, and two patients even needed an eye surgically removed [1,3]. Such cases illustrate the destructive potential of fireworks and the strain they place on medical resources in a short time frame. In Germany, consumer fireworks are strictly regulated. Smaller devices like sparklers, fountains, and small firecrackers can be purchased by adults (age ≥ 18) or adolescents (age ≥ 12 for very low-grade items), but larger explosive fireworks (category F3/F4) require special permits [1]. Consequently, powerful pyrotechnics are often obtained illegally, frequently from neighboring countries [1]. These illicit fireworks typically contain excessive explosive material and have unregulated, fast-burning fuses, making them extremely hazardous [4,5]. Improvised or homemade explosive devices are equally dangerous. The most severe firework injuries in Germany are predominantly caused by these illegal or misused high-grade fireworks, which can essentially function like small bombs. This presents a serious public health concern, as emergency departments must manage complex blast injuries in addition to the predictable seasonal increase in firework-related burn injuries around New Year’s Eve in Germany [6]. Healthcare resources are stretched by the surge of incidents. Fire departments in Berlin even triple-staff on New Year’s Eve to handle the increase in fires and injuries, and hospitals often need multidisciplinary teams (trauma, burn, hand surgery, ophthalmology) on standby [1]. The financial and logistical burden of treating multiple trauma patients simultaneously (often requiring surgery, intensive care, and rehabilitation) is significant for the healthcare system [2,6]. Notably, during the Covid-19 pandemic, Germany imposed temporary bans on the sale and public use of fireworks for the 2020/21 and 2021/22 New Year’s Eve holidays. These measures led to a dramatic reduction in firework-related injuries across the country. Compared to approximately 500 injuries in a typical year, only 79 cases were recorded during the 2020/21 New Year’s Eve and 193 in 2021/22 [7]. This represents a 75–85% reduction in acute injuries coinciding with the ban. However, there were indications of risk compensation: the proportion of incidents involving unofficial, homemade, or smuggled fireworks rose from around 3% to nearly 10% during the ban period [7]. In absolute terms, the number of illegal-firework injuries was still small, but they tended to be among the more severe cases. Overall, the bans clearly demonstrated that limiting civilian firework use can markedly reduce hospital admissions [7]. These observations provide a unique context for understanding and preventing firework injuries. In the following, we present two representative cases: a 16-year-old injured by an illegal firework and a 19-year-old injured by a homemade device—to illustrate the complex injury patterns, clinical management, and implications for prevention.

## 2. Case Presentations

### 2.1. Case 1 Presentation

A 16-year-old male suffered severe injuries while attempting to detonate an illegal Polish firework ball bomb inside a pipe on New Year’s Eve. Due to insufficient distance from the explosion, he sustained extensive trauma, including deep partial-thickness and full-thickness burns affecting approximately 9% of his total body surface area (TBSA). The burns involved his forehead, cheeks, chin, neck, and the dorsal side of his left arm and hand (Figure 1). Additionally, he sustained multiple fractures, including a slightly displaced radial styloid process fracture, an avulsion fracture of the ulnar styloid process, a triquetrum fracture, a hamate fracture, and intra-articular fractures of the third, fourth, and fifth metacarpal base. His left hand also showed extensive soft-tissue damage with compartment syndrome, necessitating urgent surgical intervention.

Management and Hospital Course:

The patient was resuscitated according to Advanced Trauma Life Support (ATLS) protocols. He received 2.5 L of crystalloid intravenous fluids and analgesia, and his wounds were covered with sterile dressings. After approximately two hours of primary stabilization at the initial hospital, the patient was transferred to our facility within the following hour. Upon arrival, he underwent secondary trauma assessment in the resuscitation bay, including urgent laboratory testing and imaging. Imaging revealed multiple fractures in the hand but no additional injuries (Figure 2). A cranial CT scan, performed due to the facial involvement, showed no intracranial bleeding or skull fractures, but revealed small, shell-like avulsion fragments at the left processus nasalis of the maxilla, as well as a diffuse galea hematoma with bifrontal and periorbital distribution, which was more pronounced on the left side. Additional hematoma formation was seen around the nose and in the left ventrolateral midface/maxillary region. The ophthalmology consult revealed no corneal or intraocular damage. A clinical neurological assessment also found no evidence of traumatic brain injury, and no further imaging was deemed necessary. Due to the severity of his burn injuries, he was emergently transferred to the severe burn intensive care unit for further management. Despite the administration of high-dose analgesia, the patient remained in severe pain. Due to the presence of circumferential burns on his left forearm, and in order to prevent subcutaneous compartment syndrome, early intervention with enzymatic debridement using Nexobrid^®^ was performed. However, progressive swelling and emerging sensory deficits in the fingers persisted despite sufficient eschar debridement, necessitating emergency surgery for compartment release. The patient subsequently underwent fasciotomy with carpal tunnel and Loge de Guyon decompression to relieve pressure and restore circulation. Although an open fracture was present, we actively decided against immediate osteosynthesis due to the massive swelling. Antibiotic therapy was initiated, but we opted to allow the hand to decongest first before considering the placement of hardware, which could have further compromised wound closure. Additionally, dermotraction was performed on the left hand after compartment release using medical skin staples and an elastic band to maintain tissue alignment and prevent retraction of tissue. Two days later, on 3 January 2025, the patient underwent a tangential necrosectomy of the nose to remove devitalized tissue, followed by application of an Epicite^®^ mask for advanced wound care. Further surgical intervention was carried out on 8 January 2025, including temporary K-wire osteosynthesis of metacarpal bones 4 and 5, along with tangential necrosectomy and debridement of necrotic tissue. Due to severe soft-tissue swelling, K-wires were used as an initial stabilization method to allow for soft-tissue recovery prior to definitive fixation. A split-thickness skin graft was harvested from the left thigh, measuring 10 × 7 cm (70 cm^2^) with a thickness of 0.2 mm, and applied to the burned areas of the left hand. The graft was meshed at a 1:1.5 ratio to facilitate drainage and flexibility. To promote optimal healing, the wound edges were carefully debrided and padding dressings were applied.

On 16 January 2025, a revision surgery was necessary due to the complexity of his fractures. This involved plate and screw osteosynthesis for metacarpal bones 4 and 5, as well as triquetrum fixation using an Aptus^®^ 2.0 system to stabilize the wrist. Throughout his hospital stay, the patient received antimicrobial therapy with Cefazolin, administered from 1 January to 8 January 2025, and again from 17 January to 20 January 2025, to prevent infection. The patient remained hospitalized for a total of 20 days, undergoing intensive physiotherapy to restore hand mobility and prevent contractures. His pain was effectively managed with an axillary plexus catheter and systemic analgesia. By the second week, he was successfully mobilized to standing and walking without complications. Upon removal of his dressings, the split-thickness skin grafts were found to be well-adhered, with only minor scarring. Despite the severity of his injuries, the patient remained fully aware of his condition and actively engaged in his rehabilitation process. His cooperation and psychological resilience contributed significantly to his recovery. He returned to work 13 weeks after the injury. Currently, the patient shows early signs of hypertrophic scarring, particularly on the dorsum of the left hand. Depending on the further course, a corrective procedure, such as Z-plasty or medical needling, may become necessary to improve scar pliability and functional outcomes. The injury, however, highlights the devastating consequences of firework-related trauma, particularly when high-explosive, illegal pyrotechnics are involved. The total hospital costs for this patient’s treatment amounted to €58,459.52, as calculated according to the standardized German DRG (diagnosis-related groups) reimbursement system, which reflects the level of clinical complexity and required resource utilization in each case [10].

### 2.2. Case 2 Presentation

A 19-year-old male sustained severe full-thickness burns to his face and both hands after a homemade firework, consisting of multiple small firecrackers enclosed in a spherical container, exploded unexpectedly. The accident occurred on 29 December 2024, at approximately 9 pm, leading to immediate and severe thermal injuries. Emergency services were alerted, and the patient was transported to the nearest hospital, where he received initial treatment, including the establishment of an arterial line and administration of 4 L of intravenous fluids. His tetanus vaccination was administered at the emergency department of the referring hospital approximately one hour after the trauma. Once stabilized, he was transferred by ground transport to our specialized burn center for further management. The patient was admitted to our facility within four hours of the trauma. The total body surface area (TBSA) affected was 10%, with 8% comprising deep partial- to full-thickness burns.

Management and Hospital Course:

The trauma and burn teams commenced resuscitation following advanced burn life support guidelines, including aggressive fluid replacement guided by the Parkland formula (with adjustments for trauma). Upon arrival in the shock room, the patient was conscious, alert, and oriented (GCS 15). His vital signs were stable without the need for catecholamine support or oxygen supply. A focused assessment with sonography for trauma (FAST) and a full-body examination revealed no additional injuries. However, significant facial swelling prevented him from opening his eyes, and intranasal soot deposition raised concerns for potential inhalation injury (Figure 3). Given the circumferential nature of the burns on both hands and the extent of facial burns, the decision was made to intubate the patient for procedural sedation and airway protection before proceeding with debridement and specialized burn treatment. Following intubation, an extensive wound debridement was performed, starting with mechanical debridement of the necrotic tissue and the application of Nexobrid^®^ to both hands and the face (Figure 4A–G).

Nexobrid^®^ was chosen due to its enzymatic selectivity, which enables precise removal of necrotic tissue while preserving viable dermal structures that are particularly important in highly functional areas such as the hands and face. This approach minimizes the risk of overtreatment and, in some cases, may help avoid the need for extensive surgical reconstruction, such as free flap coverage. Due to the severity of burns and the systemic inflammatory response, norepinephrine was required for circulatory support during the procedure. After the removal of Nexobrid^®^, the burns on both hands and the face remained extensive and deep (deep partial-thickness and full-thickness burns), necessitating further surgical intervention. On 3 January 2025, mechanical debridement was performed, followed by application of Kerecis^®^ to both hands and an Epicite^®^ mask to the face (Figure 3H,I). Kerecis^®^, a fish-skin xenograft, was selected in this case to optimize the wound bed before autografting, particularly in a setting where no immediate indication for surgical grafting was present. Its bedside application allowed wound coverage without consuming operating room resources and enabled continued tissue preparation under ICU conditions.

Moreover, the use of Kerecis^®^ is billable in the German DRG system under the OPS code for xenogeneic skin substitutes, which contributes to cost recovery within the case-based reimbursement structure. Despite initial concerns regarding possible inhalation injury, the patient’s respiratory status remained stable throughout the ICU stay. However, due to persistent and severe upper airway edema and facial swelling, the process of weaning from mechanical ventilation was prolonged. His sedation was adjusted, transitioning to isoflurane, while his ventilatory support was gradually reduced using a CPAP mode with stepwise pressure adjustments. After a prolonged weaning process, the patient was successfully extubated on 4 January 2025. Following extubation, the patient reported transient visual disturbances, prompting an ophthalmology consultation on 6 January 2025. The temporary vision impairment was attributed to periorbital swelling from facial burns. Fortunately, the ophthalmology consult found no corneal or intraocular injury, and his vision gradually returned to normal without requiring intervention. One of the key challenges in this patient’s care was managing his fluid status. Despite aggressive resuscitation, his urine output initially remained very low. Guided by PiCCO monitoring, which indicated a significantly increased need for intravascular volume, fluids were administered accordingly. This resulted in noticeable fluid overload, which we initially accepted in order to maintain hemodynamic stability in the context of capillary leak syndrome. To counter the resulting overload, we started diuretic therapy with furosemide, which led to a good response.

After extubation, the patient was transitioned to torasemide for a short period to help maintain fluid balance. Fortunately, just two days later, he was able to regulate his fluid status independently again. He received parenteral nutrition with SmofKabiven^®^, which was gradually transitioned to enteral feeding. As his oral intake improved after extubation, parenteral support was discontinued, and he was slowly reintroduced to a normal diet. By 7 January 2025, the bolster dressings on both hands were removed, revealing clean wounds with well-adhered fish-skin grafts. On the day of discharge, all surgical staples were removed, and the patient’s wounds showed satisfactory healing without signs of infection. Despite the severity of his injuries, the patient remained hemodynamically stable throughout his hospitalization. Although the patient demonstrated a stable clinical course and a favorable outcome, this case highlights the substantial financial burden associated with the treatment of severe burn injuries. The total hospital stay spanned 25 days, requiring intensive, multidisciplinary care that included invasive mechanical ventilation, multiple surgical debridements, advanced wound management utilizing Epicite^®^ mask and Kerecis^®^ grafts, parenteral nutrition, diuretic therapy, and extensive physiotherapy to prevent long-term functional impairment. The total cost of hospitalization amounted to €94.230,23. The patient returned to work 10 weeks after the injury. At the three-month follow-up, he presented with post-inflammatory hyperpigmentation and persistent redness of the burn wounds on the hand, as well as a 1 × 1 cm area of unstable scar tissue on the dorsum of the left hand. This case highlights the severe consequences of handling homemade explosive devices, particularly the consequences of high-explosive firework misuse. The pattern of injuries, including extensive deep burns and significant facial swelling, is characteristic of blast-related thermal trauma. The need for early surgical intervention, prolonged ventilatory support, and specialized wound care emphasizes the complexity of firework-induced burn injuries.

## 3. Discussion

Both cases highlight the severe, combat-like injuries that can result from improper or illegal use of fireworks, especially high-explosive or homemade devices, with patterns more typical of military or industrial blasts rather than standard consumer fireworks.

In Case 1, the patient sustained extensive hand trauma, fractures, burns, and compartment syndrome. In Case 2, the patient suffered deep burns, facial swelling, and possible inhalation injury, injuries resembling those seen with improvised explosive devices (IEDs). These patterns, involving multiple body regions and injury types, make both acute and long-term care especially challenging. A key feature of illegal high-explosive fireworks, like ball bombs, is their 360-degree blast radius, increasing the risk of widespread injuries [1]. Reports from major trauma centers indicate that the worst firework-related injuries, including hand amputations, deep burns, and loss of vision, almost exclusively involve illegally obtained or homemade fireworks [2]. In contrast, standard consumer fireworks typically cause more localized injuries, such as isolated burns to the hands or eyes, rather than the polytrauma seen in these cases [1,11].

The high-risk nature of firework injuries necessitates a multidisciplinary approach involving burn specialists, orthopedic surgeons, plastic surgeons, physiotherapists, and intensive care teams, including psychologists. In Case 1, the patient required multiple surgeries, including compartment release, fracture fixation, and extensive skin grafting, followed by a prolonged rehabilitation process. In Case 2, the need for prolonged ventilation, repeated surgical wound cleaning, and specialized treatments like fish-skin grafts and Epicite^®^ masks shows just how complex and demanding the care of firework-related burn injuries can be. But the real impact goes far beyond the intensive care unit. Because of the long time on the ventilator and the severe injuries to his hands, the patient required extensive physiotherapy just to regain basic function. Regular wound dressing changes, compression garments, and scar care became part of his daily routine. On top of the physical recovery, we must also consider the emotional side: the trauma of the explosion itself, the ICU stay, the pain, and the long healing process often leave deep psychological scars. Since the burn injuries from the explosion affected not only the skin but also caused fractures along with nerve and vascular damage, they can lead to lasting changes in how a person moves, feels, and experiences daily life. Recovery is rarely straightforward: it is a long, often painful journey that goes far beyond treating wounds. Supporting patients like this requires more than just medical interventions; it takes a dedicated, multidisciplinary team to guide them through physical rehabilitation and emotional healing alike.

Beyond the individual medical consequences, firework-related trauma places a substantial burden on healthcare systems. Severe cases like these require extended ICU care, multiple surgical interventions, intensive wound care, and advanced technologies. In such cases, healthcare resource utilization is substantial. For instance, Case 1 required almost three weeks of hospitalization, including 10 days in a specialized burn ICU, while Case 2 required 25 days in the ICU, multiple surgeries, and prolonged mechanical ventilation. Combined, the direct acute treatment costs totaled €152,689.75—€58,459.52 in Case 1 and €94,230.23 in Case 2, excluding expenses for follow-up care, rehabilitation, and potential long-term disability. The average cost for treating severe burns in Germany is approximately €32,000–40,000 for similar inpatient cases [12]. In comparison, the average inpatient treatment cost for firework-related injuries in the United States is approximately $30,000 [13]. Thus, the costs for both cases were significantly higher, reflecting the combined nature and complexity of the trauma.

Studies confirm that homemade or illegal explosive fireworks result in the highest rates of hospital admission, surgical intervention, and long-term complications, including amputations and vision loss [2,11]. These cases exemplify the extensive medical and societal costs of preventable injuries, including resource strain, prolonged rehabilitation, and indirect losses, like reduced productivity.

Notably, injury rates dropped significantly during Germany’s COVID-19 firework ban, underlining the preventive value of regulation. However, the resurgence of injuries following the ban’s lifting underscores the ongoing public health risk posed by high-explosive fireworks. Severe cases strain trauma center resources and place emergency personnel and bystanders at risk. Trauma centers, emergency services, and even bystanders are placed at risk, with young adult males most frequently affected [2,6]. Recent fatalities linked to homemade fireworks further emphasize the urgent need for enhanced public education, regulatory measures, and the promotion of professional displays as safer alternatives [1].

To build on these insights and address the ongoing threat posed by illegal fireworks, we recommend the following measures:Stricter enforcement and regulation: Close legal loopholes and increase border checks to prevent the importation of illegal fireworks. Authorities should clamp down on the sale and distribution of high-explosive fireworks to the general public, with heavier penalties for illicit supplies. This includes continuing the push for EU-wide regulations to eliminate disparities that allow “firework tourism” for stronger devices.Public awareness campaigns: Launch nationwide education campaigns about the dangers of fireworks, particularly highlighting the risks of illegal and homemade devices. Using real-life case examples, testimonials, and visuals (e.g., X-rays of blast injuries) can be effective in conveying the message.Promotion of safety and alternatives: Encourage safer celebration alternatives (such as laser light shows or officially supervised communal firework events) to satisfy cultural traditions without the same risk.Emergency preparedness: Given that some firework injuries will still occur, trauma systems should be prepared. Hospitals in high-risk areas should consider increasing staffing on peak nights and ensuring that enough burn units, hand surgeons, and ophthalmologists are on call. Prehospital providers should train for blast scenarios, and stockpile essential supplies (bandages, tetanus toxoid, antibiotics) ahead of celebrations. Ongoing surveillance of injury patterns will help adjust strategies each year.

## 4. Conclusions

These cases vividly demonstrate the devastating consequences and substantial financial burden associated with the misuse of fireworks. Key clinical insights gained from these cases underscore the importance of several critical aspects in the management of complex firework-related injuries. Foremost is the necessity for early and well-coordinated trauma care, particularly when dealing with blast injuries that affect multiple body regions simultaneously. Timely stabilization, accurate triage, and the rapid involvement of specialized burn and reconstructive teams can significantly influence both short- and long-term outcomes.

A critical takeaway from these cases is the necessity of early, coordinated trauma care, supported by hospital preparedness plans tailored for blast injuries, especially during high-risk periods, such as New Year’s Eve. Trauma system protocols, surgical readiness, and ICU capacity must be aligned with the predictable annual spike in such injuries.

In summary, firework-related injuries are largely preventable. A comprehensive, multi-tiered approach combining legislation, education, and emergency system readiness is essential to reduce the burden of these devastating injuries, protect healthcare resources, and prevent young individuals from suffering life-altering outcomes [7].

## Figures and Tables

**Figure 1 ebj-06-00031-f001:**
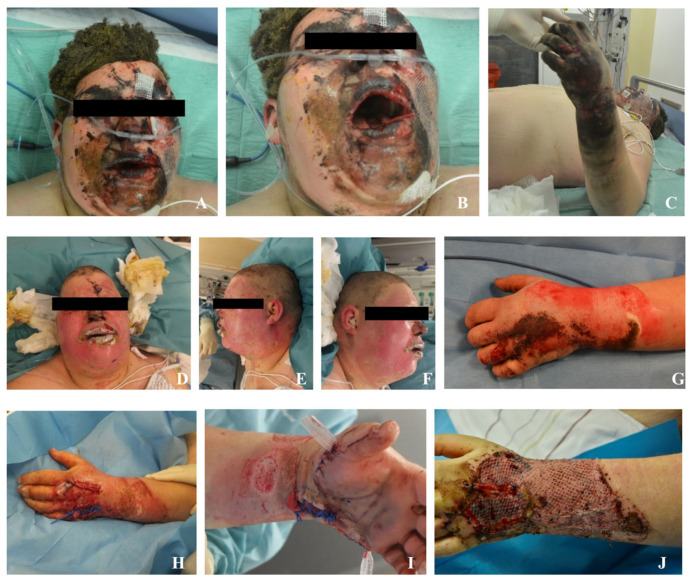
(**A**–**C**): Images showing the patient’s condition at the time of hospital admission, with severe facial and upper limb burns. (**D**–**F**): Second day after hospital admission. (**G**): After Fast-Track-Nexobrid^®^ [8,9] (**H**,**I**): Images following compartment release surgery to alleviate pressure and enhance circulation. (**J**): Post-skin-grafting, illustrating the coverage and healing of the burned areas.

**Figure 2 ebj-06-00031-f002:**
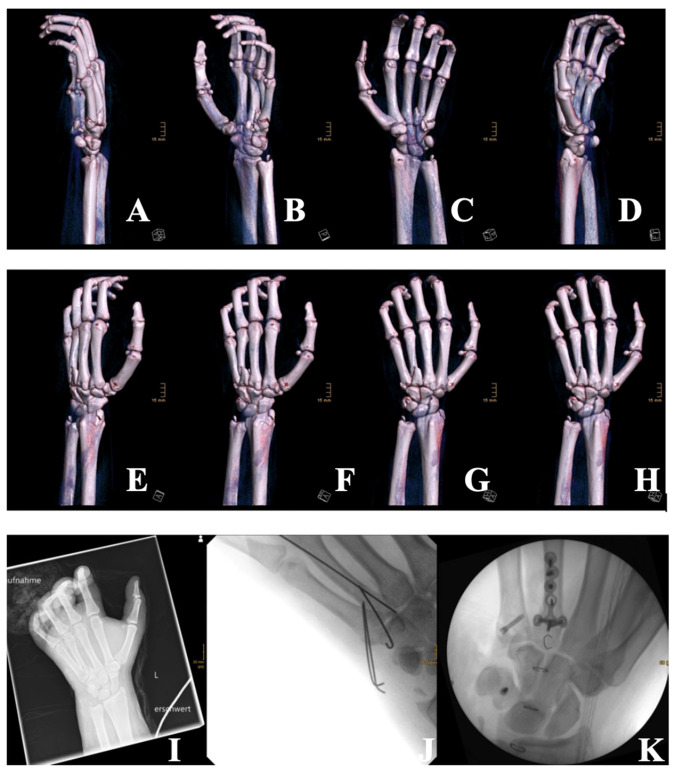
Three-dimensional CT reconstructions (**A**–**H**) taken at the time of admission and showing multiple fractures of the left hand, including metacarpal fractures (3rd, 4th, and 5th), a triquetrum fracture, a hamate fracture, and fractures of the radial and ulnar styloid processes. Extensive soft-tissue damage is also visible. (**I**): X-ray imaging of the left hand upon admission, which confirmed multiple intra-articular and styloid fractures. (**J**): Postoperative radiograph following temporary stabilization with K-wires of the metacarpal bones 4 and 5 due to extensive swelling and soft-tissue trauma. (**K**): Intraoperative fluoroscopic image after revision surgery on 16 January 2025, showing final osteosynthesis with plates and screws for the metacarpals, and triquetrum fixation using the Aptus^®^ 2.0 system.

**Figure 3 ebj-06-00031-f003:**
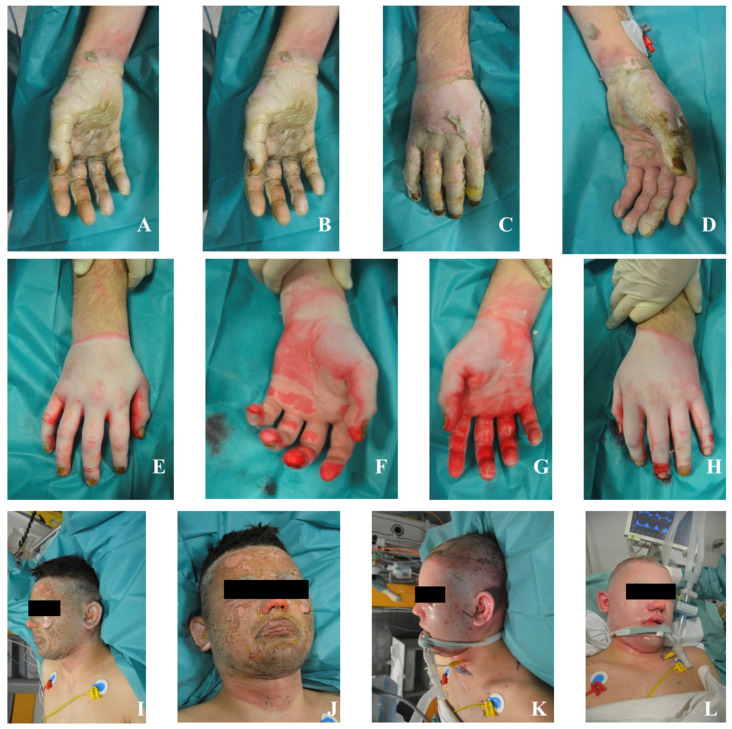
(**A**–**D**): Images showing the patient’s condition at the time of admission, with severe burns on the hands. (**E**–**H**): Images after the first debridement, illustrating deep circumferential full-thickness burns with an urgent need for treatment. (**I**,**J**): Initial facial burns upon admission, with visible signs of injury. (**K**,**L**): Post-debridement of facial wounds, demonstrating the immediate changes following the mechanical debridement.

**Figure 4 ebj-06-00031-f004:**
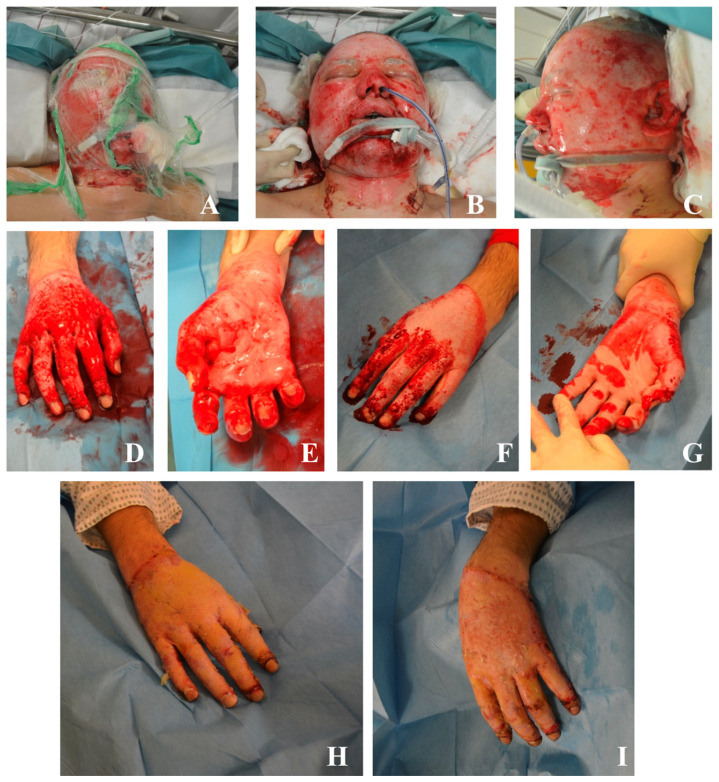
(**A**): Application of Nexobrid^®^ to the patient’s face for enzymatic debridement. (**B**–**G**): Post-Nexobrid treatment, showing the progression of wound healing and debridement results. (**H**,**I**): Kerecis^®^ grafting performed on the hands, demonstrating the coverage and improvement in skin condition following the graft.

## Data Availability

Fireworks injury statistics were obtained from current literature and reports to ensure accuracy and context. Case details were retrieved from the internal documentation systems of the treating hospital. No datasets were generated or analyzed during the current study.

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
