# Peer review of "Explosion-Related Polytrauma from Illicit Pyrotechnics: Two Case Reports and a Public Health Perspective"

_2673-1991, 2025, doi:10.3390/ebj6020031_

Round 1
Reviewer 1 Report
Comments and Suggestions for Authors
Dear colleagues!
Treatment of polytrauma has always been one of the most difficult areas of surgery and traumatology. The topic of the article is interesting because it touches on injuries sustained when using fireworks. Despite the "peaceful", everyday nature of these injuries, they largely repeat the mechanisms of explosive and gunshot wounds during armed conflicts. Victims of careless use of fireworks are often young people, teenagers, and often children. The experience of a large surgical center in organizing the treatment of this type of injury is certainly very valuable. The authors described not only the features of the clinical picture of polytrauma, but also the organization of multidisciplinary care for patients. Therefore, the contribution of the authors of the article is valuable, interesting, and should be welcomed.
At the same time, the article needs to be revised before it can be published.
- In the "Abstract" section, it is advisable to mention "combined trauma" among the keywords.
- The presentation of case â„–1 does not contain the results of the X-ray examination of the limb fractures. The combined nature and severity of the injury in patient #1 is due to simultaneous thermal and mechanical damage. Since an important advantage of the work is a detailed description of specific clinical cases, an illustration of the severity of the damage to the bones of the limbs is necessary. Please do this.
- The presentation of case 1 does not contain a description of the condition of the cornea, there is no description of the picture of the traumatic brain injury. Judging by the appearance of the facial injuries, corneal damage and traumatic brain injury are highly likely. In any case, indicating the results of the patient's examination by an ophthalmologist and a neurologist will make the description of the clinical picture more complete. Please do this.
- The presentation of case 1 (lines 93-103) describes in detail the patient's transfers between hospitals. However, there is no indication of the time it took. Please clarify how much time the patient spent in the hospital for primary resuscitation, after how many hours/days he was transferred to the burn center. 5. Lines 110-111 describe "dermotraction". How was it performed?
- Lines 116-118 describe autodermal transplantation. What is the area of the skin transplants, what is their perforation coefficient?
- The description of clinical case 1 states that the cost of treatment was €58,459 (line 134). How does this amount compare with the average cost of treating patients with burns of other etiologies? Please provide information.
- The description of clinical case â„– 2 (line 182) states that the patient was examined by an ophthalmologist, but no corneal damage was found. Please clarify how this patient was examined and what caused the temporary visual impairment.
- The description of clinical case â„– 2 does not contain data on the total area of burns and the area of deep burns. This information is needed when describing a similar clinical case. Please do this.
- The description of the 2nd clinical case states that the cost of treatment was €94,230. How does this amount compare with the average cost of treating patients with burns of other etiologies? Please let us know.
- The "Discussion" section appears twice. The one starting on line 273 is obviously an error. It needs to be corrected.
- The "Conclusions" section contains mainly reflections on the need for changes in legislation and state control over the circulation of fireworks. These reflections and recommendations, despite the fact that they are quite rational, are not a direct result of the work of the authors of the article. All of them should be moved to the "Discussion" section. The "Conclusions" section should present the results of the analysis of clinical cases, namely the features of the organization, tactics and technique of treating patients with such pathology.
To summarize, it should be said that the article makes a very good impression on the whole. It is relevant, informative, the material is well presented and analyzed. There is no doubt about the competence and extensive practical experience of the authors of the article. At the same time, it is regrettable that the valuable clinical experience in organizing the provision of care and in the direct treatment of patients is poorly reflected in the conclusions. After making the necessary corrections, the article will undoubtedly become even more informative.
Author Response
Please see the attachment.
Thank you again for your valuable feedback!

Reviewer 2 Report
Comments and Suggestions for Authors
The submitted paper points to a focalization of the severe consequences
ofthe illegally or improperly used fireworks during new year's eve
in Germany.
In the introduction paragraph Authors highlight the importance of the regulatory
laws citing a reduction in the epidemiology of fireworks injuries during a period
of temporary ban on the sales and the public use of fireworks.
Moreover a high proportion of these lesions is due to illegale fireworks.
Indeed the Case reports underline this aspect since both presented cases were injuried
by illegal or home made fireworks.
Authors state that standard consumer fireworks typically cause more localizza injuries
but they don't provide evidences about this important issue: citations or
personal data are requested.
Another important aspect of the clinical course is the treatment of the
cutaneous burns: why Authors decided to employ Nexobrid for the debridement and Epicite
in case 1 and Nexobrid, Kerecis and Epicite in case 2?
Is there a local protocol?
Moreover these products have an elevated price and could have increased
significantly the total amount of costs.
Another question: how were the costs of the treatment
determined? Authors should declare.
Moreover it seems to me that the post-dimission and recovery phases
have not been adequately described. After how many days patients returned
to work? Were the scars normal or hypertrophic?
There were permanent disabilities?
Authors should explain.
Finally some papers avalable in the literature have not been considered:
I suggest to read and comment on Alinia S.and others J.Inj. Violence Res. 2013 Jan; 5 (1); 11-16
and Gordon A.M. Orthopedics 2023; 46(3):180-184.
Best regards
y
Author Response
Please see the attachment. Thank you for your valuable feedback.

Reviewer 3 Report
Comments and Suggestions for Authors
Manuscript ID: ebj-3653428
Title: Explosion-Related Polytrauma from Illicit Pyrotechnics: Two Case Reports and a Public Health Perspective
Maria Fueth, Simon Bausen, Sonja Verena Schmidt et al.
Overview and general recommendation:
Burn injuries and thermo-mechanical trauma resulting from legal or illegal fireworks pose a significant risk to users and spectators each year. These incidents also place considerable strain on emergency services, healthcare systems, and society as a whole. This case report by Ms. Fueth and colleagues highlights two recent examples. The authors vividly and clearly present the two cases and discuss their implications in this excellently written work. Finally, they also put forward recommendations for policymakers, event organizers, and healthcare providers.
Since these are case reports, an ethics approval was not required. The patients were informed about the planned publication of their cases and gave their consent
In summary, I noticed only a few minor points that would benefit from a brief revision.
1.1 Major Comments
none
1.2 Minor Comments
- Page 1 Line 19 Please add the word “firework” to “Polish ball bomb” to create a clear distinction from tactical weapons. (Polish firework ball bomb)
- Page 1 Line 20 Specify: “[…], compartment syndrome of the forearm […]”
- Page 1 Line 25 Please add the registered trade mark “[…] Kerecis ® […] Epicite ® […]”
- Page 1 Line 26 “[…] Total hospital costs […]”
- Page 1 Line 28 What do you mean by “[…] explosive trauma remains difficult to reverse […]”. Please specify, as there is no way to reverse an explosion…not yet…Maybe you meant “difficult to treat”?
- Page 1 Line 40 What does “UKB” stand for? In the Abbreviations I can find a german text, but I had to google the translation. Why do you use an abbreviation here? Please spell out the abbreviation, as it will make it easier for the reader.
- Page 1 Line 42 Please check the reference, it does not seem to match the context of the sentence? I could not find the information about the UKB in this ref.
- Page 2 Line 54 Please explain your impressions of “in addition to routine burns every holiday season.” Is there a problem of burns during holidays in Germany? Do you got a ref for this? Please add.
- Page 2 Line 78 “Polish firework ball bomb”
- Page 3 Line 90 I had to look up the "Fast-Track Concept." If I am correct, this term was first mentioned in 2022 and describes the application of Nexobrid without the corresponding pre-soaking for fresh injuries. Perhaps providing a source reference here would be helpful for interested readers? ( doi: 10.3390/ebj3020029 )
- Page 3 Line 95 Perhaps do you know the amount of volume?
- Page 4 Lines 143 – 144 Is it truly customary in Germany to administer a tetanus vaccination immediately after a fresh injury in a pre-hospital setting? I believe you mean that the vaccination was carried out at the initial treatment clinic. Please kindly correct me if I am mistaken.
- Page 7 Line 218 Please add “rather than” instead of “than”.
- Page 9 Lines 273 – 277 This has to be erased, I think.
- Page 11 Line 355 “Coronavirus disease 2019”, “UKB – English expression, please”, “pm – post meridiem “ (small letters, please), “LD – Linear dichroism” (Where do I find it in the text?)
- Page 11 Lines 365-370 Add date of view, please
Author Response

(The authors gave the same response as above.)

Round 2
Reviewer 1 Report
Comments and Suggestions for Authors
Dear colleagues!
I have no more objections or comments. The article is interesting, relevant, well presented and illustrated. I wish you success.
Reviewer 2 Report
Comments and Suggestions for Authors
I recommend this paper for publication in the actual version